# Optimization of a Food List for Food Frequency Questionnaires Using Mixed Integer Linear Programming: A Proof of Concept Based on Data from the Second German National Nutrition Survey

**DOI:** 10.3390/nu15245098

**Published:** 2023-12-13

**Authors:** Julia Blaurock, Thorsten Heuer, Kurt Gedrich

**Affiliations:** 1Research Group Public Health Nutrition, ZIEL—Institute for Food & Health, Technical University of Munich, Weihenstephaner Berg 1, 85354 Freising, Germany; julia.blaurock@tum.de; 2Department of Nutritional Behaviour, Max Rubner-Institut, Federal Research Institute of Nutrition and Food, Haid-und-Neu-Straße 9, 76131 Karlsruhe, Germany; thorsten.heuer@mri.bund.de

**Keywords:** nutrient intake, nutrition surveys, food frequency questionnaire, mixed integer linear programming

## Abstract

Food Frequency Questionnaires (FFQs) are important instruments to assess dietary intake in large epidemiological studies. To determine dietary intake correctly, food lists need to be adapted depending on the study aim and the target population. The present work compiles food lists for an FFQ with Mixed Integer Linear Programming (MILP) to minimize the number of foods in the food list. The optimized food lists were compared with the validated eNutri FFQ. The constraints of the MILP aimed to identify food items with a high nutrient coverage in a population and with a high interindividual variability. The optimization was based on data from the second German National Nutrition Survey. The resulting food lists were shorter than the one used in the validated eNutri FFQ.

## 1. Introduction

Nutrition is one of the most crucial lifestyle factors, as it is essential to maintain health and reduce the risk of developing chronic diseases [1,2]. To assess dietary intake, dietary surveys are of crucial importance. Based on a valid assessment of dietary intake, appropriate public health measures can be derived for dietary improvement [3]. In epidemiological studies, FFQs are often used to capture subjects’ usual food intake. The questionnaire retrospectively enquires about the frequencies and portion sizes of the foods consumed in a previous time period (e.g., a week or a month) using a predefined food list [4].

Due to the reduced burden of response (low effort and time to complete), and cost-efficiency, FFQs are particularly useful for groups that are difficult to reach and for large epidemiological studies [5]. Therefore, FFQs provide an important contribution to public health research.

The main objectives of FFQs are [6,7]:The assessment of the usual long-term dietary intake;The sufficient assessment of the nutrient intakes of interest to sufficiently reflect the dietary intake of a target group;The ranking of the participants based on their food and nutrient intakes to enable the comparison of disease risk among the different dietary intake levels.

To fulfill these requirements, the food items of an FFQ should be selected based on the following criteria [6]:Regular consumption (or at least within the time span of the FFQ);A high coverage of nutrient intakes;Large interindividual variability to depict differences in the dietary intake between persons.

For optimal usability, compliance, and validity, the food list should be as short as possible but as comprehensive as necessary to meet the FFQ requirements [8]. Short food lists improve usability by reducing the time needed to complete the questionnaire and increasing response rates [9]. 

The food lists and portion sizes in the FFQ differ depending on the study population, region, and study goal (e.g., nutrient, food-specific, or generalized dietary survey) [10]. To validly assess dietary intakes, FFQs need to be adapted to these three parameters requiring a modification of the food list and portion sizes, which are often queried using photos of different portion sizes.

The compilation and adjustment of food lists for FFQs are often based on either quantitative population-based dietary intake data or qualitative data from focus group interviews [6].

Dietary intake data can be used to identify food items with the highest contribution to the overall intake level or to identify food items that account for a certain amount of interindividual variation. For the latter, stepwise regression is commonly used. In this approach, food items are progressively added or removed based on their ability to explain variation in the intakes of the nutrients of interest. The process stops when a specific level of explained variance (R^2^) is reached [11,12].

Focus group interviews are often used to adapt food lists, e.g., for a specific population, to collect qualitative information about participants’ dietary behaviors and food choices [13]. 

However, these methods do not minimize the number of food items. Consequently, inefficient food lists may result.

To minimize the number of foods in the food list and to identify food items that contribute to the coverage of the nutrients of interest as well as to the variance of nutrients, an optimization based on mixed integer linear programming (MILP) can be conducted [14]. In the optimization model, a binary decision variable defines whether food items should be included in an FFQ.

In this work, a MILP was used to generate food lists for a general dietary survey in Germany based on dietary intake data from the second German National Nutrition Survey [15]. 

The aims of the present paper are as follows:(a)To minimize food lists with mixed integer linear programming;(b)To compare the number of selected food items as follows:-Across the different proportions of nutrient coverage and of variance coverage;-Across the different nutrients of interest;-With the existing eNutri FFQ, which is designed to assess a generalnutrient intake [16];(c)To analyze the correlation between the nutrient coverage and variance coverage;(d)To identify and analyze the optimal subsets of food items for energy intake and compare them with the existing eNutri FFQ.

## 2. Materials and Methods

### 2.1. Study Population and Dietary Intake Data

Data from the latest representative nutrition survey in Germany was used for optimization. The second German National Nutrition Survey collected consumption data between 2005 and 2007 from subjects in Germany aged between 14 and 80 years [17].

For the present analysis, two 24 h recalls (24 HR) from 13 926 participants were used to calculate an average daily nutrient intake per person and food item. Within this subset were 7669 women and 6257 men; the mean age was 45.8 years (Table 1). 

24 HR were collected by trained interviewers using the software EPIC-Soft(NVS II version). Participants were initially asked about the food and beverages they had consumed in the previous day, which were then specified in more detail in the following process. For portion size estimation, photos of portion sizes were used in addition to household measures and standard units. The second 24 HR was conducted on average 15 days after the first recall [18,19]. 

Dietary intake data were available per person and per food item. The average daily nutrient intake per person was calculated using the German food composition database (German Nutrient Database, Bundeslebensmittelschlüssel, BLS version 3.02). 

Two optimizations with different food aggregation levels were conducted. Aggregation levels were derived from the food classification system of the BLS. In the BLS, each food item is assigned a unique 7-digit code which classifies food items hierarchically into food groups and subgroups. 

For the first optimization, 184 BLS food subgroups were used, corresponding to the first two digits of the BLS (aggregation level 1).

For the second optimization, the first three BLS digits were used, which describe foods at the individual food item level, resulting in a total of 1908 food items (aggregation level 2). Food groups or food items that were not consumed were disregarded.

If a person did not consume one of the 184 (1908) foods, their consumption was quantified with 0 g to avoid a bias in the calculation of the variance. The 184 (1908) foods used for optimization will be referred to as “food items” in the following. 

### 2.2. eNutri FFQ

The food list created with MILP was compared with the existing eNutri FFQ version 2.0. eNutri is an online FFQ that comprises 156 food items and has been used and validated in various studies assessing habitual food intake [16,20].

In the eNutri FFQ, participants first select whether and how often they consumed individual food items in the previous month (e.g., never, 1x, 2–3x per day, etc.). Then, participants are asked to indicate their typical portion size for each food item by selecting one of three portion size photos on the screen. For some food items, there are additional questions that allow for a more precise differentiation (e.g., full-fat or low-fat yoghurt), aiming at increasing the questionnaire’s validity.

### 2.3. Optimization and Comparison of FFQ Length

MILP was used to minimize the number of food items in the food list for the two different aggregation levels. MILP is an optimization method that combines linear programming and integer programming. This method is suitable for selecting an optimal number of food items for an FFQ, as the decision variables *x_n_* for the food item *n* must take integer values [14].

Given a set of foods, *N*, and a set of nutrients, *J*, the optimization problem can be formulated as follows:(1)$$minimize\;\sum_{n=1}^{N}x_{n\;}\;$$

for $x_{n}$ ∈ {0,1}, *n* = specific food item for *n* ∈ {1, …, 184} for aggregation level 1, and *n* ∈ {1, …, 1908} for aggregation level 2$x_{n}$ is a binary decision variable indicating whether food item *n* is in the food listif $x_{n}$ = 1, food item *n* is includedif $x_{n}$ = 0, food item *n* is not included
(2)$$\begin{array}{c} subject\;to\sum_{n=1}^{N}x_{n}\;\cdot \;C_{1,n}\;\geq \;b\;\\ \vdots \\ \;\sum_{n=1}^{N}x_{n}\;\cdot \;C_{j,n}\;\geq \;b\\ \vdots \\ \sum_{n=1}^{N}x_{n}\;\cdot \;C_{40,n}\;\geq \;b \end{array}$$
for *j* ∈ {1, …, 40}, *j* = specific nutrientfor *b* ∈ {0.6, … 0.99}, b = arbitrary threshold $C_{j,n}\;$= percentual contribution of food item *n* to the overall intake of nutrient *j*
$$C_{j,n}=\frac{\;q_{j,n}}{\sum_{n=1}^{N}q_{j,n}}$$$q_{j,n}=$ total intake of nutrient *j* from food item *n* over all persons
(3)$$\begin{array}{c} and\;\;\;\;\;\;\sum_{n=1}^{N}x_{n}\;\cdot \;{\;S}_{1,n}\;\geq \;b\\ \vdots \\ \sum_{n=1}^{N}x_{n}\;\cdot \;{\;S}_{j,n}\;\geq \;b\\ \vdots \\ \sum_{n=1}^{N}x_{n}\;\cdot \;{\;S}_{40,n}\;\geq \;b \end{array}$$
$$Var(q_{n,j})=\frac{1}{I}\sum_{i=1}^{I=13,926}\left(q_{j,n,i}-\bar{q_{j,n}}\right)2$$
for *i* ∈ {1, …, 13,926}, *i* = specific person$q_{j,n,i}=\mathrm{in}\;\mathrm{take}\;\mathrm{of}\;\mathrm{nutrient}\;j\;\mathrm{from}\;\mathrm{food}\;\mathrm{item}$  *n* by person *i*$S_{j,n}$ = percentual contribution of food item *n* to the sum of variances of the overall intake of nutrient *j*
$$S_{j,n}=\frac{Var(q_{n,j})}{\sum_{n=1}^{N}Var(q_{n,j})}$$

The optimization involves one objective function to minimize the number of food items and two sets of constraints, both comprising 40 nutrient specific constrains, i.e., j ∈ {1, …, 40}. The two sets of constraints reflect two previously mentioned requirements of food items in an FFQ: high nutrient coverage and large interindividual variability of nutrients. The algorithm iteratively searches for the combination of food items that minimizes the number of items while satisfying the two sets of constraints.

The nutrient coverage constraints aim to select food items that account for a large proportion of the intake of the selected nutrients in the target population. The constraints ensure that the percentual coverage for selected nutrients is greater or equal to a specific threshold value b.

The variance coverage constraints aim to select food items that account for a large interindividual variance in nutrient intakes. To determine food items with a high contribution to variance, the conventional approach would involve using R^2^. However, there are two challenges in applying R^2^ in this context. First, R^2^ cannot be formulated as a linear equation, which is necessary for MILP. Second, the contribution of a food item to R^2^ is not explicitly determined but depends on the food items already included in the model. To address these challenges, food items were selected based on their percentual contribution to the sum of variances of a specific nutrient. These variance coverage constraints ensure that the percentual contribution of selected food items to the sum of variances of the overall intake of nutrient j is greater or equal to a specific threshold, b.

To compare optimal food sets across different sets of nutrients, the number of nutrients in the optimization was incrementally increased. Overall, a maximum of 40 nutrients that comprehensively reflect a general diet were considered in the optimization. First, optimization for energy intake was conducted, then the intakes of carbohydrates, protein and fat were added and eventually, the set of nutrients considered was extended with 36 vitamins and minerals. All nutrients included in the optimization are listed in Appendix A Table A1. 

To analyze the development of the number of food items, MILP was conducted for different threshold values, b. For both sets of constraints, the same values for b were used. The initially chosen value of b was 0.60, which gradually increased by 0.05 until 0.95 was reached. The final optimization was run with b = 0.99. 

Calculations were conducted with the statistical software R version 4.3.0. Optimization was solved using the R package ROI 1.0-1, which provides an interface for various solvers. In this case, the solver GLPK was used. 

To test which of the two sets of constraints required more food items, the optimization was additionally performed with only one of the sets of constraints, respectively. The resulting food lists were compared regarding the number and the kind of food items. It was assumed that food items with a large nutrient coverage also have a large coverage of the sum of variances. Therefore, Pearson’s correlation coefficients between nutrient coverage and variance coverage were calculated for energy, carbohydrates, protein, and fat intakes.

To identify the relative change in the number of optimal food items, growth rates were calculated with
$$\frac{N_{1}-N_{0}}{N_{0}}$$
where $N_{0}$ is the number of food items resulting from MILP with a lower proportion of both nutrient coverage and variance coverage. $N_{1}$ is the number of food items resulting from MILP with a 5% higher proportion of both nutrient coverage and variance coverage.

### 2.4. Optimal Sets of Food Items

To showcase concrete food lists, optimal sets of food items were identified for selected proportions, b, for energy intake at aggregation level 1. Due to the nature of MILP, there might be more than one set of foods meeting all the conditions specified above. To determine the optimal food set among several solutions, the objective function was extended by a penalty term, ensuring that the resulting set of foods provides the greatest coverage in both nutrient intake and the sum of variances:(4)$$minimize\;\sum_{n=1}^{N}x_{n}-C_{j,n}\;\cdot \;x_{n}-S_{j,n}\cdot \;x_{n}$$

This term penalizes the inclusion or exclusion of food items, *x_n_*, based on their coverage, $C_{j,n}$, for nutrient *j* (in this case energy intake) and based on their coverage for the sum of variances, *S*_*j*,*n*_, within nutrient j (energy intake).

Food sets with a minimal number of food items across all levels of energy intake coverage and energy intake variance were compared in terms of their included food items.

## 3. Results

### 3.1. Dietary Intake Data

The mean energy intake from the 24 HR was 1980 kcal/d (sd = 734 kcal/d). Non-consumption applied to 19 food items of all 184 of the food items in aggregation level 1. The food items that were not consumed are listed in Appendix A Table A2.

The food lists generated with MILP differ regarding the level of nutrient coverage, variance coverage, and selected nutrients.

### 3.2. Optimization and Comparison of FFQ Length

The optimization was conducted for the two aggregation levels of the food groups defined by the BLS. 

Figure 1 shows the number of optimal food items for aggregation level 1 with 184 food groups. For 60% coverage of the energy intake and variance of energy intake, the minimum number of food items required was 23. The number of selected food items increases continuously: for 95% and 99% coverage of the energy intake and variance of energy intake, 76 and 111 food items were selected, respectively. 

Considering all 40 nutrients, the selected food items additionally increased for each proportion of nutrient coverage and variance coverage: For 60% nutrient coverage and nutrient variance, 38 food items were selected. For 95% and 99% nutrient coverage and variance coverage, 97 and 128 food items were required, respectively.

Compared with the eNutri FFQ, which includes 156 food items, the food list generated with MILP contained fewer food items across all the proportions of nutrient coverage and variance coverage.

Even when considering all 40 nutrients, only 128 food items are needed to account for 99% of the nutrient coverage and variance coverage.

Considering the energy intake, the MILP conducted separately with the nutrient coverage constraints and the variance coverage constraints showed that the nutrient coverage constraints were consistently binding. The variance coverage constraints were non-binding, as indicated by the lower number of food items (Figure 2a). For example, for b = 0.95, the nutrient coverage constraints require 76 food items, whereas the variance coverage constraints require only 68 food items.

Considering all 40 nutrients, the number of food items resulting from the MILP conducted separately with the nutrient coverage constraints and the variance coverage constraints coincide (Figure 2b). For b = 0.95, the nutrient coverage constraints require 94 food items and the variance coverage constraints require 96 food items.

The Spearman correlation coefficients, calculated over food items, between the proportion of nutrient coverage and the proportion of variance coverage were calculated for the intakes of energy, carbohydrates, protein, and fat and ranged from 0.812 to 0.957, indicating a strong positive linear association between the nutrient coverage and variance coverage (Table 2). This implies that food items with a high nutrient coverage also tend to have high coverage in a nutrient’s sum of variance. Additionally, all the correlation coefficients were statistically significant, with *p* < 0.001 resulting from the t-tests comparing the correlation between the nutrient coverage and variance coverage against the null hypothesis of no correlation.

The growth rates show the percentual growth of the number of food items compared to the previous level (Figure 3). Up to b = 0.9, the growth rates ranged between 0.1 and 0.2. For example, for all 40 nutrients with b = 0.8, the growth rate amounts to 0.13, meaning that the number of food items increased by 13% compared to b = 0.75. 

For higher levels of nutrient coverage and variance coverage, the growth rates increased non-linearly up to 0.46.

The growth rates were generally higher for the food lists that only considered energy intake, carbohydrates, protein, and fat than for the food lists that considered all 40 nutrients.

The minimal number of required food items for aggregation level 2, using 1908 food items, is generally higher than the minimal number of food items at aggregation level 1, as food items at aggregation level 2 are at the individual food item level. The minimal numbers of food items at aggregation level 2 for energy intake ranged from 62 (b = 0.6) to 459 (b = 0.99) and for all 40 nutrients from 98 (b = 0.6) to 497 (b = 0.99). The growth pattern of the food items at aggregation level 2 was similar to the one observed at aggregation level 1. 

The detailed results for aggregation level 2 are in Appendix A Figure A1 and Figure A2.

### 3.3. Optimal Sets of Food Items

For the energy intake at aggregation level 1, the minimal food set, which simultaneously provides the largest coverage of nutrients and the largest coverage of the sum of variances, was identified. A detailed breakdown of the food items for the selected proportions for the coverage of the energy intake and variance of energy intake is provided in Appendix A Table A3.

The food items were classified into broader food categories to describe the food lists generated with the MILP and to compare the results with the food list of the eNutri FFQ. For this purpose, the hierarchical food classification system of the BLS was used to differentiate the 20 major food categories (Table 3). 

With 60% nutrient coverage and coverage of sum of variance, four food items were from the main BLS food category *bread and rolls*, making it the largest food category. It was followed by three food items from the category *milk, dairy products, and cheese* and by two food items each from the categories *cakes, tarts, pastries and biscuits*; *non-alcoholic beverages*; *alcoholic beverages*; *oils and fats*; *sweets and sugar;* and from the category *sausages and other meat products*. One food item came from each of the categories *fruit and fruit products*; *potatoes and potato products*; *meat;* and from the category *composite dishes mainly containing vegetable products*.

With higher proportions of nutrient coverage and variance coverage, the food set is extended by additional food groups. For a 99% coverage of the energy intake and variance of energy intake, 111 food items from 19 major food categories were included. 

The eNutri FFQ comprises 156 food items from 20 major food categories. The largest food category was *vegetables and vegetable products,* with 25 food items. In contrast, the food lists generated by the MILP for 60% and 80% of the energy intake and variance of energy intake did not comprise any food items from this category. Furthermore, the eNutri FFQ used seven food items from the category *deep-sea and fresh-water fishes, shellfish,* whereas all the MILP-generated food lists disregarded this category.

Compared to eNutri, the number of food items in the MILP-generated food list was consistently smaller, even up to 99% nutrient coverage and 99% variance coverage. Compared to eNutri, the food lists were 85% to 29% shorter.

## 4. Discussion

Compared to the validated online FFQ eNutri, the food list generated by the MILP was shorter for all the scenarios. With 128 food items, 99% nutrient coverage and 99% coverage of the sum of variances could be achieved for all 40 nutrients, in contrast to the eNutri FFQ, which comprises 156 food items. 

The number of required food items increased with both the number of nutrients considered as well as the proportions of the nutrient coverage or variance coverage. Almost ubiquitous nutrients such as macronutrients require a relatively large number of food items to satisfy the required nutrient intake and nutrient variance compared to the minerals or vitamins that were considered among those 40 nutrients. For example, if only the energy intake is considered, as many as 42 food items are needed to reach b = 0.8, and only 19 food items have to be added to the food list such that all the 40 nutrients are adequately taken into account. One reason could be that specific nutrients like certain vitamins are concentrated in only a few food items, leading to a smaller number of foods needed to capture their intakes. These results were consistent with other studies [21,22]. For example, Shai et al. found that only three food groups explained 80% of the variance in the consumption of vitamin E.

Since energy intake, however, is basically distributed across all foods, more food items are needed to explain the total energy intake and variance in energy [21].

The growth rate of the food items at aggregation level 2 using 1908 food items showed a similar pattern as observed with the 184 food items at aggregation level 1. Including a broader range of food items allows to capture finer details and nuances within the data and a more precise representation of the food items consumed. Performing the optimization with a larger number of food items demonstrates the ability of the optimization algorithm to handle large-scale datasets efficiently. This scalability is an advantage of MILP compared to other methods for selecting food items.

For separate optimization with only one set of constraints, respectively, fewer food items were needed to explain the variance in the energy intake between persons than to explain the total energy intake. This difference disappeared completely as the number of considered nutrients increased to 40. Other studies found a higher difference between the total nutrient intake and nutrient variance [22,23]. For example, Kim et al. found that 119 food items could explain 90% of the total energy intake, whereas only 71 food items were needed to explain 90% of the variance between persons [23]. 

Optimizing an FFQ involves setting appropriate proportions for the nutrient coverage and variance coverage. The present paper assessed a range from b = 0.60 to b = 0.99 for the two sets of constraints. In practice, the choice of proportion can be adapted considering the study aim and time constraints of future study participants.

The threshold might influence usability in terms of the FFQ length. A higher threshold results in a bigger number of food items, leading to a longer questionnaire and potentially greater burden of response [9]. 

Other studies developing FFQs used a threshold of at least an 80% contribution to the nutrient intake and at least an 80% explanation of the between-person variability [21,24].

The MILP identified a unique solution for the minimum number of food items. However, there are several options of methods to compile the food items into minimal sets. Due to the nature of the MILP, there might result several lists with a minimal number of food items, all meeting the defined constraints. Therefore, a penalty term was added to the objective function to identify the optimal food set that explains the greatest coverage and variance for the energy intake subject to the given constraints. The resulting food list primarily consisted of foods with a high energy content, as the main purpose was to explain the energy intake. Food items with a lower explanatory content regarding the energy intake, such as vegetables, were omitted from the list. In the optimization model, the focus was to explain the nutrient intake, while an explanation of the dietary behavior played a secondary role.

Due to the marginal differences between other food lists regarding the coverage of the energy intake and variance of energy intake, and due to the inclusion of other factors such as eating behavior, the identified food list’s relevance in real-world applications should be further evaluated. However, this paper provides a mathematical framework to deduce a food list for an FFQ. Apart from the compilation of new food lists, this framework could help researchers to assess whether an existing food list aligns with their study aims (e.g., whether the food list satisfyingly covers the selected nutrients). This can enhance the validity and reliability of research findings.

The optimization for the present study was conducted using dietary intake data from the German National Nutrition Survey from 2005 to 2007. Unfortunately, no more recent data on the dietary intake in the German population are available. This database might not reflect the current preferences in nutrition, and consequently, the current dietary trends might not be covered by the food items selected using the MILP. However, the MILP can easily be rerun with new dietary data as soon as those become available. Additionally, there are further advantages associated with using the MILP compared to other methods like focus groups or stepwise regression to compile food lists: first, with appropriate underlying dietary intake datasets, food lists can easily be adapted to study target, dietary habits, or other population groups (e.g., other countries) [14].

Second, food lists are reproducible when using the same underlying datasets, leading to reliable results [14]. Third, optimization techniques are scalable, as they can handle large-scale datasets with numerous food items and persons. 

With the MILP, only the selection of food items was considered. However, the accuracy of FFQs is influenced by various other factors. For example, the way foods are grouped can affect the validity of the answers: Thompson et al. showed that asking about multiple related food items in a single question resulted in less underreporting than asking about foods in multiple separate questions [25]. 

Also, the order in which food items are presented in an FFQ can influence the accuracy of the reported food intake. Querying specific food items before general food items can lead to a higher reported food intake [25].

For the optimization, it was assumed that food items are uncorrelated. Yet, some food items are likely to be dependent entities, as they might rather be consumed in combinations, for example, as part of a meal [26]. The dependencies among food items might lead to inaccuracies concerning their contribution to the sum of variances. However, this simplification was made to reduce the complexity of the optimization problem. Furthermore, the resulting list should be regarded as a minimum set of food items and might be supplemented with additional considerations to improve the accuracy and completeness of the food list. The results of this paper suggest that the number of food items required to assess the dietary intake in a population might be far less than in the currently available FFQs.

Another major strength of this paper is that the MILP is based on a large dietary intake dataset with people from all age groups. In addition, the dietary intake was recorded using multiple 24 HRs, which provide good validity in dietary assessments [27].

## 5. Conclusions

MILP offers a data-driven, reliable and comprehensive approach to selecting food items for an FFQ. Generating a food list by means of a MILP demonstrates that the number of food items could be reduced compared to the previously used eNutri FFQ. The resulting food lists from the MILP can be easily adapted to the dietary habits of different population groups and different study objectives. The identified food sets provide a basis which could be supplemented by further considerations from nutritional experts. Furthermore, an MILP can help researchers to assess the appropriateness of an existing food list regarding their study aim.

## Figures and Tables

**Figure 1 nutrients-15-05098-f001:**
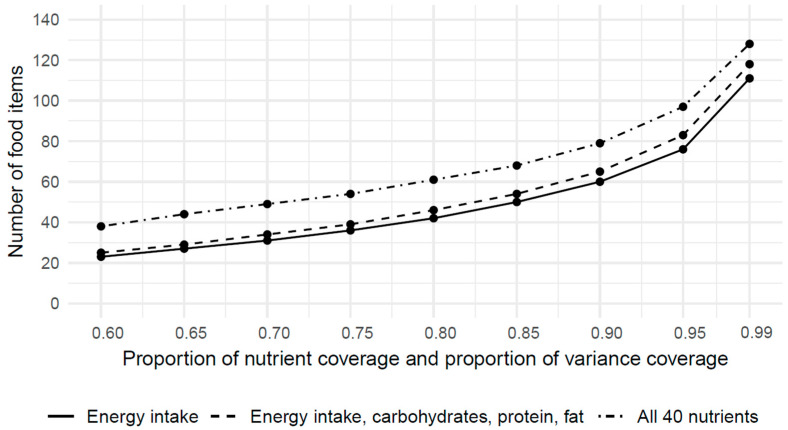
Minimal number of required food items on aggregation level 1, calculated with mixed integer linear programming.

**Figure 2 nutrients-15-05098-f002:**
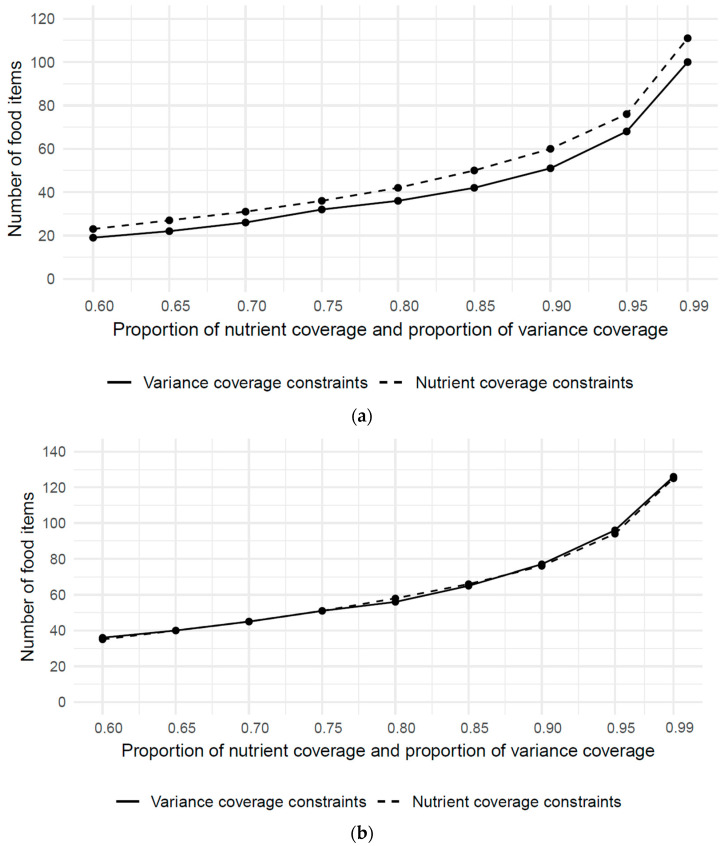
(**a**). Minimal number of required food items, with respect to nutrient coverage constraints and variance coverage constraints, considering only energy intake. (**b**). Minimal number of required food items, with respect to nutrient coverage constraints and variance coverage constraints, considering the intake of 40 nutrients.

**Figure 3 nutrients-15-05098-f003:**
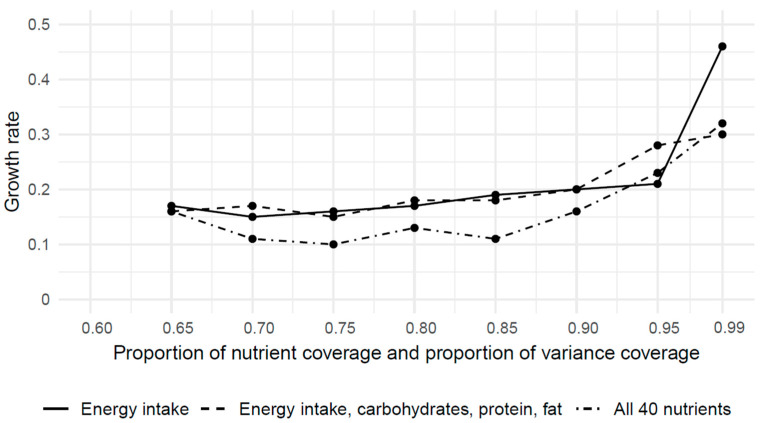
Growth rates of the minimal number of required food items, with respect to nutrient coverage constraints and variance coverage constraints (aggregation level 1).

**Table 1 nutrients-15-05098-t001:** Characteristics for the second German National Nutrition Survey (*n* = 13,926).

		Overall *n* (%)
Sex		
	Male	6257 (45)
	Female	7669 (55)
Age group (years)		
	14–30	2983 (21.4)
	31–45	3830 (27.5)
	46–60	3654 (26.2)
	>60	3459 (24.8)

**Table 2 nutrients-15-05098-t002:** Pearson’s correlation coefficients between proportions of nutrient coverage and proportions of variance coverage for different nutrients.

Nutrient	Pearson Correlation Coefficient (95% Confidence Interval)	*p*-Value ^1^
Energy intake (kcal)	0.902 (0.870; 0.925)	<0.001
Carbohydrates	0.939 (0.919; 0.954)	<0.001
Protein	0.812 (0.757; 0.856)	<0.001
Fat	0.957 (0.942; 0.967)	<0.001

^1^ *p*-values resulting from *t*-test comparing the correlation between proportions of nutrient coverage and proportions of nutrient variance against the null hypothesis of no correlation.

**Table 3 nutrients-15-05098-t003:** Food lists for different proportions of energy intake and variance of energy intake, compared to eNutri food list.

	Number of Food Items Selected by MILP for Major Food Categories			eNutri FFQ v.2.0
Major BLS Categories	Proportion of Energy Intake and Proportion of Variance of Energy Intake			
b	0.60	0.80	0.99	
Total number of food items	23	42	111	156
Bread and rolls	4	3	7	9
Milk, dairy products, cheese	3	3	8	11
Cakes, tarts, pastries, biscuits	2	5	7	8
Non-alcoholic beverages (coffee, tea, soft drinks)	2	4	5	4
Sweets and sugar	2	3	8	7
Alcoholic beverages (beer, wine, spirits)	2	2	6	5
Oils and fats	2	2	7	7
Sausages and other meat products	2	2	6	6
Composite dishes containing mainly vegetable products	1	4	9	15
Meat (excluding organs) beef, veal, pork, mutton	1	3	6	4
Fruit and fruit products	1	3	7	13
Potatoes and potato products, starchy roots and tubers, mushrooms	1	1	3	5
Eggs and egg products, noodles		1	4	2
Cereal products, grains, flours, milled products, rice		1	3	6
Venison, poultry, feathered game, offal		1	1	4
Composite dishes containing mainly animal products			7	10
Vegetables and vegetable products			6	25
Legumes (mature), pulses, nuts, oil- and other seeds			3	6
Spices, seasonings, raising agents, condiments			2	2
Deep-sea and fresh-water fishes, shellfish				7

## Data Availability

Data available on request due to restrictions eg privacy or ethical.

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
