# Peer review of "Optimization of a Food List for Food Frequency Questionnaires Using Mixed Integer Linear Programming: A Proof of Concept Based on Data from the Second German National Nutrition Survey"

_nutrients, 2023, doi:10.3390/nu15245098_

Round 1

Reviewer 1 Report

Comments and Suggestions for Authors

The aim of the research is very useful as short FFQ is more efficient and easier for researchers for dietary assessment studies if this tool includes common nutrients availability. I have the following suggestions/comments: 

·       It is true, as it mentioned in the abstract, that any “food lists need to be adapted depending on the study aim and the target population”.  Could this Short FFQ tool be used for diverse demographic population? 

·       The FFQ with Mixed Integer Linear Programming (MILP) uses fewer food items in the food list. It would be helpful to explain clearly how many food (%) are excluded? And what factors and how this exclusion determined in the abstract? Basically, how the optimization was taken? 

·       Some of the nutrients, such as starch, which is very general, and maltose, which is not that common for nutrient assessment, included in the appendix 1. Explain the rationale behind the food items in optimization. 

·       In the methods indicated that the sources of data were Dietary Intake Data and German National Nutrition Survey. Does then the product, short FFQ, more applicable for German population only? 

·       It would be helpful to explain about the implementation, maybe limitations, of this tool for other countries such as low income and developing counties (practicality and usage of this tool for other counties). 

·       The MILP and then Pearson correlation to determine the coefficient between nutrient coverage and variance were done very meritoriously.

This paper demonstrates the more efficient usage of FFQ tool.  

Comments on the Quality of English Language

The quality of English language is fine, just a few minor editing and additions to clarify the ideas. Overall, a careful editing is suggested. 

Reviewer 2 Report

Comments and Suggestions for Authors

Dear Authors,

I consider your study innovative, well-structured and it proves an extensive experience gathered in the field of food frequency questionnaires. Your manuscript topic is original and mixed integer linear programming may be a viable alternative method for tailoring food frequency questionnaires to different targeted groups.

To enhance the accessibility for readers, since a higher number of food items leads to an extended questionnaire and most probably to errors, please explain in a detailed and comprehensible manner how MILP can minimize the number of food items.

Please explain also why composite dishes, vegetables, legumes, spices, and fish are included only for the 99% proportion of energy intake.

Another possible question is related to the impact of eating behaviors on the minimum identified food list proposed by MILP.

The results should be valued more in the discussion and conclusions sections. Most of the selected references are adequate and up to date for the chosen topic.

All the best wishes!

Comments on the Quality of English Language

There are required few minor editing corrections (English U.K. vs. English U.S.A.)
